# Demodulation Technique Based on Laser Interference for Weak Photo-Acoustic Signals on Water Surface

**Xiaolin Zhang \*, Hongjie Mao and Wenyan Tang**

Institute of Precision Instruments, Harbin Institute of Technology, Harbin 150001, China;
xiaomaolv2hao@163.com (H.M.); tangwy@hit.edu.cn (W.T.)

**\*** Correspondence: zhangxiaolin@hit.edu.cn; Tel.: +86-139-3626-1257

**Abstract:** To detect underwater sound-generating targets, a water surface acoustic wave laser interference and signal demodulation technique is proposed in this paper. The underlying principle of this technique involves casting a laser beam onto the water surface disturbed by an underwater acoustic source and creating interference between lights reflected by the surface and reference lights. A data acquisition and processing system was employed to obtain water surface acoustic wave information from the interference signals by means of demodulation, thus allowing detection of the underwater target. For the purpose of this study, an interference detection platform was set up in an optical dark chamber. High-frequency water surface fluctuations were introduced in the reference optical path as the phase generated carriers to create laser interference signals in two different paths, which received demodulation based on an improved arc tangent demodulation algorithm and characteristic ratio algorithm, respectively, in view of their different frequencies. Water surface wave information was then derived from such low-frequency and high-frequency signals. According to test results, in the frequency range of 200 Hz–10 kHz, the frequency detection accuracy was better than 1 Hz. The amplitude measurements exhibited high repeatability, with a standard deviation lower than 2.5 nm. The theory proposed in this paper is therefore experimentally verified with good results.

**Keywords:** optical interference; phase generated carrier; phase demodulation; water surface acoustic waves

## 1. Introduction

When an acoustic wave originating from an underwater object reaches the water/air interface, it will generate an elastic surface wave that transmits transversely along the interface in the medium's superficial layer, with a penetration depth of one wavelength approximately [1]. It has been discovered through studies that if an underwater acoustic source causes water surface vibration, only those vibrations sharing the same frequency as the acoustic source are significant in amplitude [2]. Hence, detection of the water surface acoustic waves (WSAWs) resulting from acoustic waves provides an indirect way of obtaining sound-generating information of an underwater acoustic source.

Currently, sonar remains the main technology for underwater sound field detection, but the detection coverage is limited by low-speed movement of vehicles. Researchers have therefore been highly interested in laser-based WSAW detection techniques [3,4]. One of the earliest studies with this regard was reported in the paper of Lee M. S. [5], which illustrated the mechanism of WSAWs stemming from underwater acoustic signals. In his experiments, he also obtained optical signals with the same frequency as their acoustic source. Since then, a number of techniques have been brought forward by other researchers, including those based on laser scattering, laser diffraction, luminous flux, and laser interference [6,7], all contributing to direct processing of light intensity signals. By using a laser sonar technique measuring Doppler vibrations, Antonelli L. T. et al. [8,9] successfully

detected WSAWs, and later completed single-point detection at different locations of dynamic water surface. Miao Runcai et al. [10] conducted laser interference measurements of acoustic waves on a low-frequency liquid surface and obtained modulation interference patterns. Information like surface acoustic wavelength, frequency, and amplitude was then demodulated from such interference patterns. Great efforts have been made by our project team in the research of underwater target detection through aerial laser interference and effective extraction of complicated underwater sound field signals, and so far, concrete results have been achieved under laboratory conditions (in the optical dark chamber with a high SNR). Detection of underwater acoustic sources in the frequency range of 1 kHz–18 kHz has been made possible by us with techniques such as spectrum analysis, wavelet transform analysis, and local turning point data demodulation [11–13]. However, all the above techniques fail to cover the whole acoustic wave frequency range, and are unable to provide information on variations of acoustic source signal density. In contrast, the new interference demodulation method proposed in this paper enables acquisition of underwater acoustic source vibration information and tracking of varying vibration density based on detection of WSAW. The effectiveness of the method has been proven by us through tests.

## 2. Materials and Methods

### 2.1. General Concept

An interference system is used for water surface detection, and information on water surface waves is contained in the phase of the detection signals. The disturbances have two components: natural disturbances of the water surface and WSAW. By expressing the water surface disturbances with simple harmonic vibration superimposition, we depict the interference signals in the following equation:

$$U(t) = A \cos\{\frac{4\pi}{\lambda}[\sum_i A_i \cos(\omega_i t + \theta_i) + A_s \cos(\omega_s t + \theta_s)] + \Omega\} \tag{1}$$

where $A$ is the system gain of the interference signals; $\lambda$ is the laser wavelength; and $\Omega$ is the fixed phase caused by the interference optical path. $A_i$, $\omega_i$, and $\theta_i$ represent amplitude, angular frequency, and initial phase of the natural disturbances waves, respectively, and $i$ represents number of harmonic waves. Similarly, $A_s$, $\omega_s$, and $\theta_s$ are amplitude, angular frequency, and initial phase of the WSAW, respectively. In Appendix A, the above equation receives a trigonometric function expansion and a Bessel function simplification in turn, followed by a simulation at an underwater acoustic source frequency of 1 kHz.

The resulting signal spectrum of Table 1 is shown in Figure 1. The dense spectral lines in the low-frequency zone are attributed to natural fluctuations of water surface and various low-frequency disturbance waves. Concentrated frequency bands exist around 1 kHz, i.e., the acoustic source frequency, and exhibit a symmetric distribution. Spectral lines also appear around multiples of 1 kHz such as 2 kHz and 3 kHz, although with negligible amplitude. In reality, the wide low-frequency bands imply superimposition of low frequency acoustic source signals and low-frequency natural water surface fluctuation signals, making it impossible to perform spectrum analysis of the underwater low-frequency signals directly. A new phase demodulation method is therefore required to obtain the vibration information.

Two demodulation algorithms were used in our study for interpretation of signals acquired in the course of underwater acoustic source detection, as shown in Figure 2. The signals provided by the photodetector underwent a DC filtering process, and a spectrum analysis was then performed on the interference signals. Different demodulation schemes were adopted depending on the frequency. For underwater high-frequency acoustic source signals above 2 kHz, frequency bands independent of the low-frequency ones were created in the spectrum, with a symmetrical distribution around the underwater sound-generating source frequency. After determination of the underwater source

frequency by locating the distribution center, a characteristic ratio method was used to obtain the WSAW amplitude. For low-frequency signals below 2 kHz, given their aliasing with low-frequency bands, we were not able to derive further information from the spectrum. To address this, an improved arc tangent function algorithm was employed to demodulate the phase generated carriers (PGCs) signals, thus characterizing phase changes caused by WSAWs, and obtaining amplitude and frequency of the WSAWs.

**Table 1.** Simulation parameter settings.

|  | Amplitude/nm | Frequency/Hz | Initial Phases/rad |
| --- | --- | --- | --- |
|  | 1500 | 3 | $0.8\pi$ |
| Disturbances (superimposition) | 3000 | 4 | $0.2\pi$ |
|  | 1000 | 5 | $1.2\pi$ |
| WSAW | 30 | 1000 | $0.5\pi$ |
| Laser wavelength $\lambda$/nm | 632.8 | | |
| System gain of the signals $A$/V | 1 | | |
| The fixed system phase $\Omega$/rad | 10,000 | | |
| SNR of system/dB | 20 | | |

WSAW = water surface acoustic wave; SNR = signal to noise ratio.

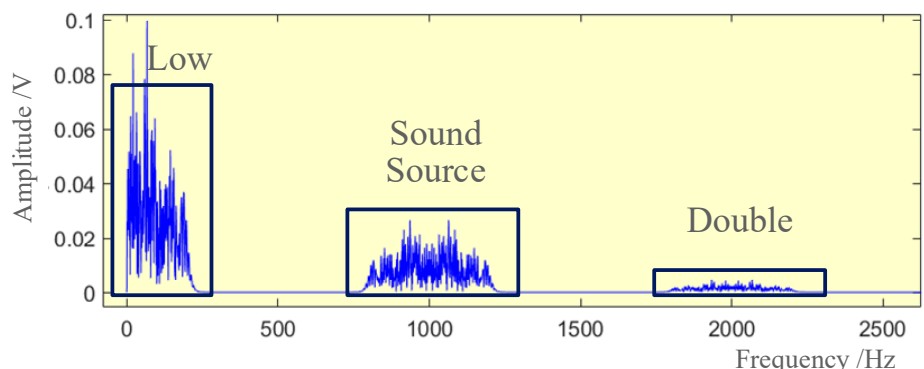

**Figure 1.** Spectral distribution of 1 kHz simulated interference signals.

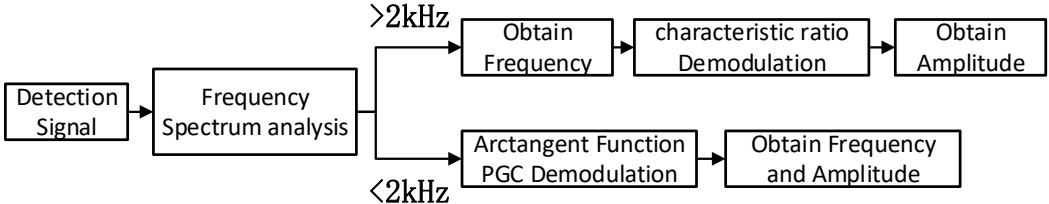

**Figure 2.** Principle diagram of demodulation algorithm for underwater acoustic source detection.

### 2.2. Characteristic Ratio Demodulation Algorithm

The spatial phase modulation and time phase modulation provided by water surface waves for incident laser are analogous. Spatial modulation depth can be estimated from the light intensity distribution of diffraction. For laser interference detection, we can also estimate the laser time phase modulation depth caused by water surface waves from the spectrum of the interference signals [14]. To achieve this, Bessel function was used to expand interference signals, and $P(\omega)$ was used to represent the amplitude of the frequency of $\omega$ in the spectrum. The characteristic ratio $R$ was then defined as:

$$R = \sqrt{\frac{\sum_m P[(2m+1)\omega_i]}{\sum_m P[(2m+1)\omega_i + \omega_s]} \cdot \frac{\sum_m P(2m\omega_i)}{\sum_m P(2m\omega_i + \omega_s)}} = \frac{J_0\left(\frac{2\pi}{\lambda}A_s\right)}{J_1\left(\frac{2\pi}{\lambda}A_s\right)} \tag{2}$$

where $(2m+1)\omega_i$ and $2m\omega_i$ were odd and even frequency doubling of low frequency components; $J_0$, $J_1$ are Bessel functions. Since the amplitude $A_i$ of water surface disturbance waves was significantly higher than WSAW amplitude $A_s$ caused by the underwater acoustic source, according to the nature of Bessel function of the first kind, $R$ can be rewritten as:

$$R = \frac{\sum_m P(m\omega_i)}{\sum_m P(m\omega_i + \omega_s)} \tag{3}$$

It can be seen from the above equation that without aliasing of the low-frequency bands with the signals under measurement, the characteristic ratio $R$ can be determined by calculating the ratio of the sum of low-frequency band amplitudes to that of the frequency component amplitudes after a frequency shift by $\omega_s$. With the characteristic ratio $R$ thus obtained, WSAW amplitude $A_s$ can be resolved with the Bessel function. Since the analytic solution of $A_s$ is hard to derive in practice, we estimate the numerical solution of $A_s$ from the look-up table which is shown in Figure 3.

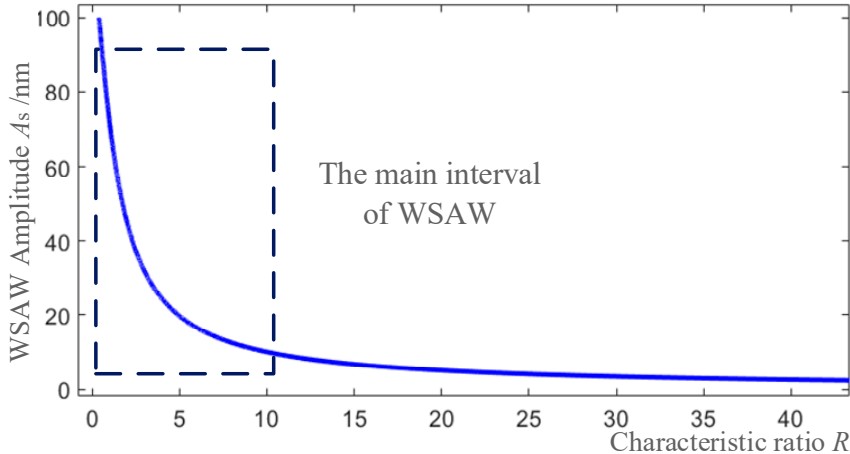

**Figure 3.** The function image of $A_s(R)$.

### 2.3. PGC-Based Improved Arc Tangent Demodulation Algorithm

When the signal frequency under test is lower than 2 kHz, PGCs are used to detect the underwater acoustic source. The test system built for this purpose is shown in Figure 4. A high-frequency underwater acoustic source was placed on the reference optical path, and the high-frequency water surface waves excited by it were introduced into the system as the carriers. This introduction of carriers can avoid the use of high-price optical phase modulators and are easy to realize; meanwhile, the interference effect is better because the reference light has a similar light intensity with the measured light. Furthermore, there are also environmental disturbances on the water surface of the reference pool, which can be reduced with measurement pool disturbances and the anti-interference ability of the system can be improved.

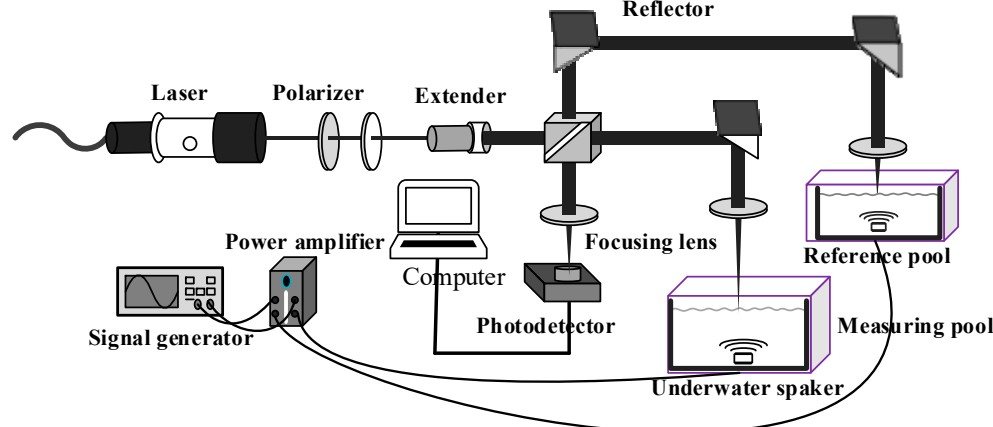

**Figure 4.** Test system diagram.

A high-frequency signal was added to the reference arm, its amplitude, frequency, and initial phase being represented by $A_c$, $\omega_c$, and $\theta_c$, respectively.

$$\begin{cases} H = \frac{2\pi}{\lambda} A_c \\ \theta(t) = \frac{2\pi}{\lambda} [\sum_i A_i \cos(\omega_i t + \theta_i) + A_s \cos(\omega_s t + \theta_s)] \end{cases} \tag{4}$$

Assuming that the reference optical path carriers have a modulation depth of $H$, and that the phase caused by water surface disturbance and WSAWs under measurement is $\theta(t)$, we achieved the following interference signals of the PGCs:

$$U(t) = A \cos[H \cos(\omega_c t + \theta_c) + \theta(t) + \Omega] \tag{5}$$

Based on Equation (5), a demodulation was performed with the configuration given in Figure 5. This process starts with a sine/cosine mixing of the carrier frequency, followed by low-pass filtering that generated three mixing signals:

$$\begin{cases} U_{1s} = -AJ_1(H) \sin(\theta_c) \sin[\theta(t)] \\ U_{1c} = -AJ_1(H) \cos(\theta_c) \sin[\theta(t)] \\ U_{2c} = -AJ_2(H) \cos(2\theta_c) \cos[\theta(t)] \end{cases} \tag{6}$$

It can be seen from the above equation that $U_{1s}$ and $U_{1c}$ signals only differ in amplitude. Their amplitude ratio is equal to the tangent of the initial phase $\theta_c$ of the carriers. Initial phase of the carriers can be derived from these two signals, which can be written as:

$$\theta_c = \arctan(\frac{U_{1s}}{U_{1c}}) \tag{7}$$

With the PGC demodulation technique, an arc tangent function demodulation method [15] was used to achieve electronic mixing, filtering, and phase division of the signals expressed by Equation (6). On this basis, the water surface fluctuations under measurement can be obtained by means of demodulation and expressed as follows:

$$\theta(t) = \arctan[\frac{J_2(H) \cos(2\theta_c)}{J_1(H) \cos(\theta_c)} \cdot \frac{U_{1c}}{U_{2c}}] \tag{8}$$

A combination of the carrier modulation depth $H$ obtained from the spectrum and the carrier initial phase $\theta_c$ in Equation (6) gives complete information on demodulated water surface

fluctuations, which, after elimination of low-frequency disturbances through high-pass filtering, provides WSAW information.

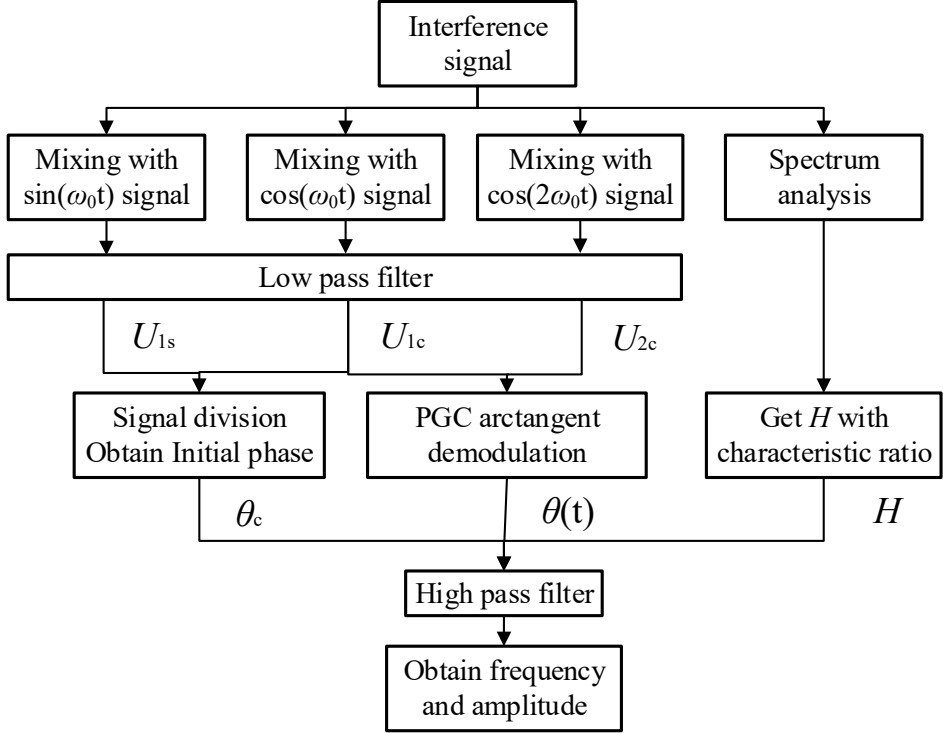

**Figure 5.** Phase generated carriers (PGCs) interference signal demodulation flow.

## 3. Results

In our study, a test platform was set-up with separate optical elements in an optical dark chamber. The optical dark chamber, interference optical path, and other test devices used in the study are shown in Figure 6. During the test, the signal generator created signals on two paths. The 10 kHz high-frequency signals on the first path acted on the reference optical path to generate high-frequency carriers of the same frequency, while the signals on the second path caused the speaker in the water pool to generate waves as the underwater target to be measured. The main equipment in our experiment is shown in Table 2.

**Table 2.** Performance index of main equipment.

| Equipment | Parameter | Value |
|---|---|---|
| He–Ne Laser | Wavelength/nm | 632.8 |
| | Power/mW | 0.6–1.4 |
| | Power stability | ±0.1% |
| Underwater speaker | Rated power/W | 15 |
| | Frequency response /Hz | 80–20,000 |
| Data acquisition card | Sampling rate/kS/s | 100 |
| | Resolution/bit | 16 |

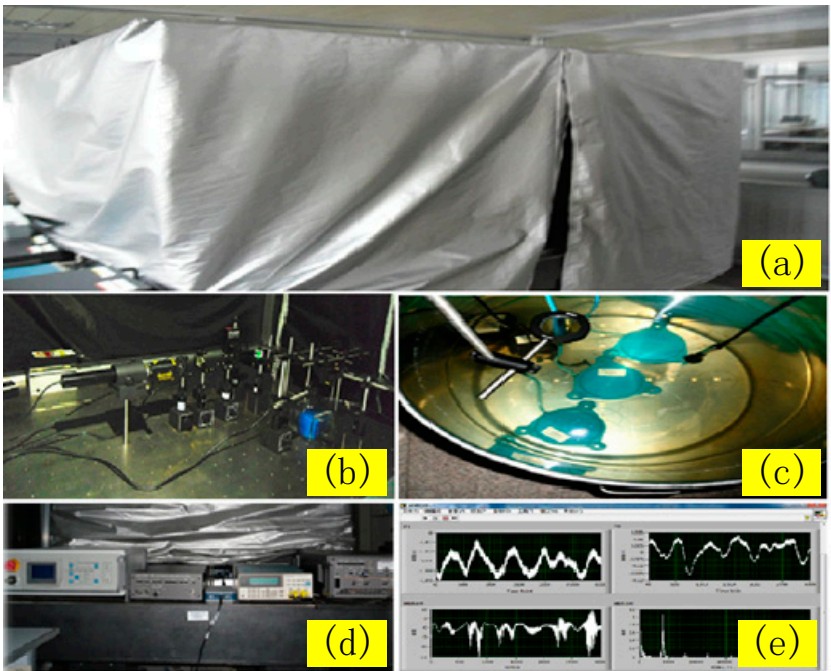

**Figure 6.** (**a**) Optical dark chamber; (**b**) interference optical path; (**c**) test water pool (height 100 cm, diameter 80 cm); (**d**) signal amplifier, power amplifier, and laser controller; (**e**) PC measurement screen.

*3.1. Frequency Measurement Results*

Taking the interference signals of 3 kHz and 1 kHz underwater acoustic sources as an example, the spectrograms are shown in Figure 7. It can be seen from this figure that the low-frequency band caused by the natural water surface fluctuations is about 1 kHz in width. The 1 kHz underwater acoustic source signals experienced aliasing with low-frequency bands in Figure 6b, which coincides with theoretical analysis.

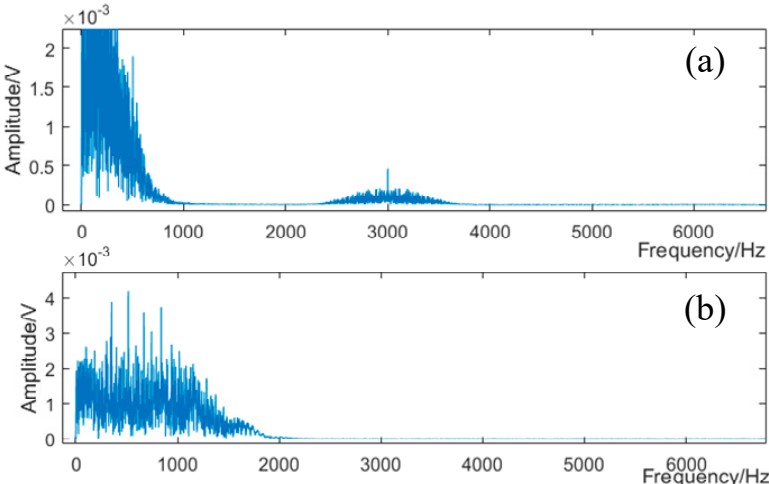

**Figure 7.** Spectrum aliasing phenomenon: (**a**) spectrum of 3 kHz source; (**b**) spectrum of 1 kHz source.

Based on the previously mentioned demodulation theory, we first grouped the detected signals in the interference signal spectrum into high-frequency and low-frequency categories. Low-frequency signals are handled with improved PGC demodulation technique. The phase demodulation signals have exactly the same frequency as the signal generator. For high-frequency signals, the central points of the high-frequency components were extracted for frequency measurement, as shown in Figure 8.

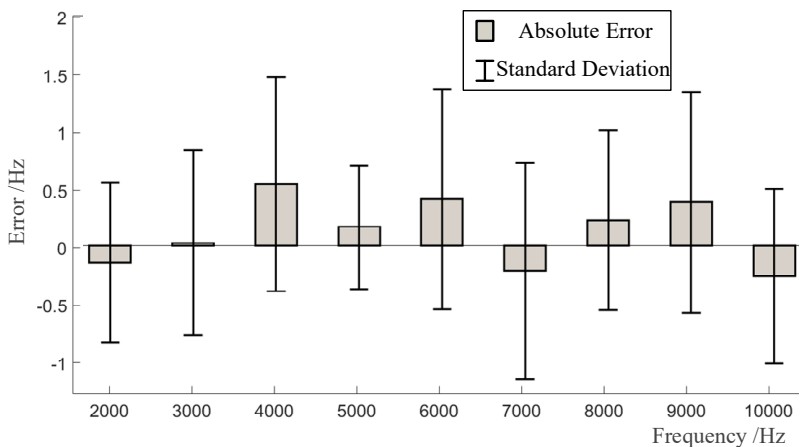

**Figure 8.** Errors based on extraction of central frequency points in the spectrum.

With the central frequency extraction method, detection tests were performed on underwater acoustic sources in the frequency range between 2 kHz and 10 kHz. For each frequency, the measurement was repeated five times. It can be seen from the results given in the above table that for signals above 2 kHz, the frequency measurements were generally accurate and repeatable, with a measurement error and a standard deviation not exceeding 1 Hz.

### 3.2. Amplitude Measurement Results

The spectrum of interference signals was analyzed first. The amplitude was derived directly with characteristic ratio method at high frequencies. In the case of low frequencies, high-frequency carriers were added to the reference optical path, and a signal generator was used to create 10 kHz high-frequency signals, which were fed to the reference speaker in the pool to excite carrier waves. This was followed by demodulation with improved PGC demodulation method. The test procedure covers all frequency bands between 200 Hz and 10 kHz, each corresponding to five repeated measurements. Some results are given in Table 3. For 500 Hz low-frequency signals, the modulation depth $H$ was first estimated from the spectrum, and then the initial phase was obtained with arc tangent method, finally leading to the amplitude demodulation results. For 3-kHz high-frequency signals, after extraction of low-frequency bandwidth, the sum of spectrum amplitudes was calculated. Next a ratio of this sum to that of spectrum amplitudes after a left frequency shift by $\omega_s$ was calculated, which is known as the characteristic ratio $R$. In the last step, the water surface acoustic wave amplitude was determined with the look-up table $A_s(R)$.

**Table 3.** Some amplitude measurement results.

| $f$/Hz | $H$ ($A_c$ = 39.28 nm) | $\theta_c$/rad | $A_s$/nm | $\sigma$/nm |
|---|---|---|---|---|
| | | 0.628 | 48.59 | |
| | | 1.309 | 50.03 | |
| 500 | 0.39 | 0.211 | 51.77 | 1.63 |
| | | 0.785 | 53.42 | |
| | | 1.553 | 51.22 | |

| $f$/Hz | Sum of Low/V | Sum of high/V | $R$ | $A_s$/nm | $\sigma$/nm |
|---|---|---|---|---|---|
| | 3.157 | 1.044 | 3.023 | 31.64 | |
| | 2.684 | 0.861 | 3.117 | 30.78 | |
| 3000 | 2.551 | 0.854 | 2.986 | 32.00 | 1.31 |
| | 4.003 | 1.408 | 2.843 | 33.44 | |
| | 3.996 | 1.453 | 2.750 | 34.44 | |

The standard deviations of amplitudes measured at different frequency bands are given in Figure 9. The figure implies a lower standard deviation of the demodulated signals at high frequencies. For low-frequency signals, due to errors arising from estimation of carrier characteristic ratio and PGC demodulation, the repeatability was lower, but still within the limit of 2.5 nm. In sum, the amplitude measurements in our study provide satisfactory repeatability.

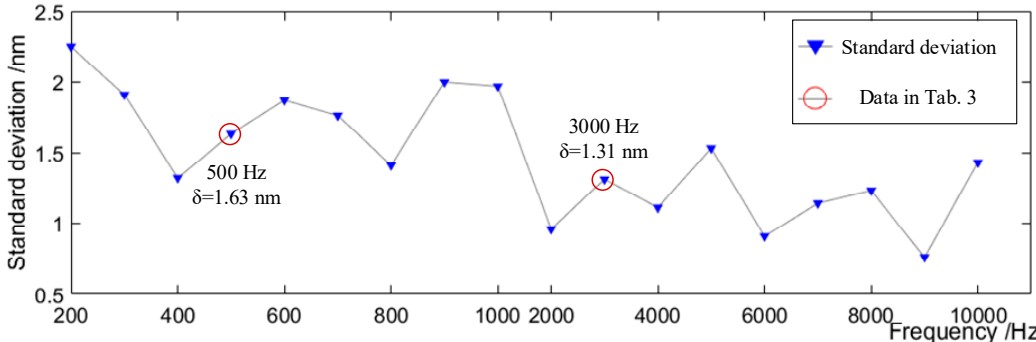

**Figure 9.** Standard deviation of amplitude measurement.

### 3.3. Results of Tracking Demodulation of Amplitude Modulation Signals

The signal generator creates 1 kHz signals with an amplitude modulated by 1 Hz sine waves. The modulation results at a water surface vibration sampling rate of 100 kS/s are given in Figure 8. Apparent periodical changes of the amplitude can be seen from the time domain signals after demodulation, and the correct 1 kHz demodulation result can be found in the spectrum analysis as well. The envelope of the time domain signals was extracted and fitted as a curve, resulting in the amplitude curve shown in Figure 10. The fitting curve equation is expressed as $A_s = -26.39 \sin(2\pi \times 1.03t - 3.59) + 65.28$. The fact that the fitting curve frequency 1.03 Hz exhibits the same as the amplitude-modulating frequency 1 Hz of the signal generator within a small tolerance proves the ability of the method used in our study to accurately track signal amplitude changes.

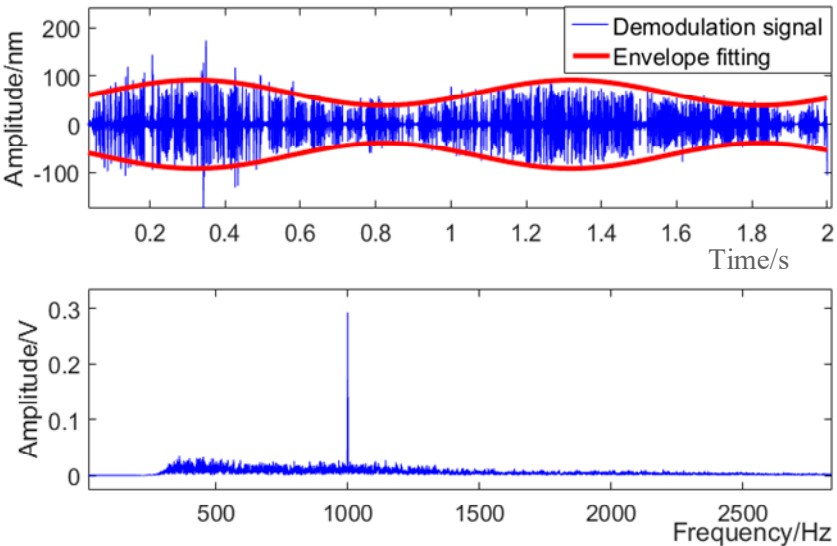

**Figure 10.** Demodulation result and fitting curve of amplitude modulation signal.

## 4. Discussion

To address spectrum aliasing that occurs during underwater acoustic source detection with laser interference, a demodulation method capable of measuring multiple frequency bands is proposed in this paper. Our method achieves lower frequency detection and tracking of change signals,

which cannot be realized by previous research. In our approach, the spectrum of interference signals was analyzed, and different solutions were chosen depending on frequencies: for frequencies higher than 2 kHz, a characteristic ratio method was adopted, while for those lower than 2 kHz, a PGC-based improved arc tangent function demodulation method was used. In the former, the central point of a high-frequency band was used as the measurement frequency of the detection signals, and characteristic ratio $R$ was calculated before the amplitude was obtained in a lookup table. The PGC-based demodulation method enables demodulation with an improved arc tangent function, which gives initial phase and modulation depth of high-frequency carriers. After low-frequency filtering, amplitude and frequency of WSAW can be extracted in the optical dark chamber (SNR about 14 dB). The detection approach adopted in this paper applies to the frequency range between 200 Hz and 10 kHz while providing a frequency measurement accuracy better than 1 Hz. It also features high repeatability for amplitude measurement, with a standard deviation not exceeding 2.5 nm. As demodulated signals are generated through demodulation, time-varying signals may be measured as well. The results of amplitude modulated signal tests show that this approach is able to track signal changes.

Certain restrictions still exist in the approach adopted in this paper. The lower limit of frequency detection is dependent on filter performance and water surface disturbance waves, due to which the demodulation effect for signals below 200 Hz is not significant. On the other end of the spectrum, the upper limit of frequency detection is related to the high-frequency carriers, and should be limitless in theory. As an underwater speaker was used in our study to excite WSAWs as carriers, the frequency response attribute of the speaker also plays a role. Frequency shift devices will therefore be incorporated in subsequent research, and will hopefully enable carriers with a frequency much higher than that of acoustic waves. Furthermore, the next experiment is to consider the impact of natural light in the natural environment, perhaps considering optical filters and other devices to improve system SNR.

**Author Contributions:** Conceived the Method and Wrote the Paper, Z.-X.L.; Designed and Performed the Experiments, M.-H.J.; Edited the Manuscript, T.-W.Y.

**Funding:** This study was funded by the National Natural Science Foundation of China (61108073); Shanghai Aerospace Science and Technology Innovation Foundation (SAST2015029); Postdoctoral Researchers Settled in Heilongjiang Research Foundation (AUGA4120006016).

**Conflicts of Interest:** The authors declare no conflict of interest.

## Appendix A

The interference signal we set in the article is:

$$U(t) = A\cos\{\frac{4\pi}{\lambda}[\sum_i A_i\cos(\omega_i t + \theta_i) + A_s\cos(\omega_s t + \theta_s)] + \Phi\} \tag{A1}$$

In order to simplify the derivation, we set $k = 2\pi/\lambda$, $\theta_i = 0$, $\theta_s = 0$, therefore:

$$U(t) = A\cos\{2k[A_i\sin(\omega_i t) + A_s\sin(\omega_s t)] + \Phi\} \tag{A2}$$

We first use the trigonometric function to expand it:

$$
\begin{aligned}
U(t) \quad &= A\cos\Phi\{\cos[2kA_i\sin(\omega_i t)] \cdot \cos[2kA_s\sin(\omega_s t)]\} \\
&= -A\cos\Phi\{\sin[2kA_i\sin(\omega_i t)] \cdot \sin[2kA_s\sin(\omega_s t)]\} \\
&= -A\sin\Phi\{\sin[2kA_i\sin(\omega_i t)] \cdot \cos[2kA_s\sin(\omega_s t)]\} \\
&= -A\sin\Phi\{\cos[2kA_i\sin(\omega_i t)] \cdot \sin[2kA_s\sin(\omega_s t)]\}
\end{aligned}
\tag{A3}
$$

Combining the expansion formula of Bessel function:

$$\begin{cases} \cos[x\sin(\omega t)] = J_0(x) + 2\sum\limits_{k=1}^{\infty} J_{2k}(x)\cos(2k\omega t) \\ \sin[x\sin(\omega t)] = 2\sum\limits_{k=0}^{\infty} J_{2k+1}(x)\sin[(2k+1)\omega t] \end{cases} \tag{A4}$$

The $J_k(x)$ is the first-class Bessel function value of $k$ order of $x$:

$$J_k(x) = \sum_{m=0}^{\infty} \frac{(-1)^m}{m!\Gamma(m+k+1)}\left(\frac{x}{2}\right)^{2m+k} \tag{A5}$$

We further simplify the interference signal by using the Bessel function. We can clearly see the frequency components and their magnitude of the interference signal:

$$\begin{aligned} U(t) \; &= A J_0(2kA_s) J_0(2kA_i)\cos\Phi \\ &+ 2A J_0(2kA_s)\cos\Phi \sum_{m=1}^{\infty} J_{2m}(2kA_i)\cos(2m\omega_i t) \\ &- 2A J_0(2kA_s)\sin\Phi \sum_{m=0}^{\infty} J_{2m+1}(2kA_i)\sin[(2m+1)\omega_i t] \\ &+ 2A J_0(2kA_i)\cos\Phi \sum_{n=1}^{\infty} J_{2n}(2kA_s)\cos(2n\omega_s t) \\ &- 2A J_0(2kA_i)\sin\Phi \sum_{n=0}^{\infty} J_{2n+1}(2kA_i)\sin[(2n+1)\omega_s t] \\ &+ 4A\cos\Phi[\sum_{m=1}^{\infty} J_{2m}(2kA_s)\cos(2m\omega_i t)]\cdot[\sum_{n=1}^{\infty} J_{2n}(2kA_i)\cos(2n\omega_s t)] \\ &- 4A\cos\Phi\left\{\sum_{m=0}^{\infty} J_{2m+1}(2kA_i)\sin[(2m+1)\omega_i t]\right\}\cdot\left\{\sum_{n=0}^{\infty} J_{2n+1}(2kA_s)\sin[(2n+1)\omega_s t]\right\} \\ &- 4A\sin\Phi[\sum_{m=1}^{\infty} J_{2m}(2kA_i)\cos(2m\omega_i t)]\cdot\left\{\sum_{n=0}^{\infty} J_{2n+1}(2kA_s)\sin[(2n+1)\omega_s t]\right\} \\ &- 4A\sin\Phi\left\{\sum_{m=0}^{\infty} J_{2m+1}(2kA_i)\sin[(2m+1)\omega_i t]\right\}\cdot[\sum_{n=1}^{\infty} J_{2n}(2kA_s)\cos(2n\omega_s t)] \end{aligned} \tag{A6}$$

There is no detailed explanation in this paper, but we describe the interference signal with simulation, which is consistent with the theory from the simulation result.

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
