# Peer review of "Demodulation Technique Based on Laser Interference for Weak Photo-Acoustic Signals on Water Surface"

_applsci, doi:10.3390/app8122423_

Round 1

Reviewer 1 Report

The authors present a novel signal demodulation technique for interferometric measurements of underwater sound fields. The work is interesting and relevant.

Some minor issues could be improved:

- In Figure1 it would be intersting to zoom out the high frequency components (or maybe use a logarithmic scale), since they can hardly be distinguished from the line of the axis.

- The characteristic ratio should be explained in more detail, in order to bettwer gasp its significance. Also, the parameter m is not explained in Equ.2. J0 and J1 probably refers to the Bessel functions, but this should clearly be stated in the text.

- In the characteristic ratio based approach, the amplitude is estmated using a look-up table. The authors should give some details about this look-up table. What are the relevant parameters? How was the table generated?

-Figure3: The scheme as well as the experimental details should be given in more detail.

- The authors should aso comment some more, on the practical relevance of their results, which prove the principle quite nicely. However, extension to a measurement in a realistic environment seems not trivial to me.

- Some typos.

Author Response

Dear Reviewer:

We appreciate the valuable suggestions and comments from you. We studied them very carefully and tried our best to revise our manuscript. Detailed revision and explanations are in the file.

Reviewer 2 Report

This paper presents a Demodulation Technique Based on Laser Interference for Weak Photo-acoustic Signals on Water Surface.
It is an interesting paper; however, I have some comments written below.

Abstract: acronym WSAW is used without writing water surface acoustic waves.
Always remember to include a spacing between a number and unit, 10kHz should be 10 kHz

Keywords: Include: Optical interference, water surface acoustical waves

The introduction is OK however a discussion of prior art with respect to signal to noise is lacking.

Materials and methods:
Eq. 1 directly describes interference signal from a perfect Michelson interferometer without describing the background of using this signal. One should start describing the surface vibrations + noise and thereafter use this signal as the phase modulation of the optical signal (two-beam interference). Consider use an illustrating figure.

Page 2 line 71: This sentence is not understood: The above formula receives a trigonometric function expansion and a Bessel function simplification in turn…..??

Figure 2: How is this figure generated (simulation parameters are missing)
Page 3 line 101: This sentence is not understood: Spatial modulation depth can be estimated from the light intensity distribution of different diffraction lights in their spatial frequency distribution pattern??

Page 3 line 105: What is precisely meant with P (amplitude of components in spectrum…)?
Page 4 line 116: The look-up table….? Not at all understood, please describe

Figure 3: provide more describing figure text (self explaining!).
Page 5, line 137, write the eq. tan(Theta)=U1s/U1c. In general: use equation instead of formula!

Figure 5: Nice with pictures, however all images are too small and blurred. Please improve significantly

Figure 6: the vertical axis has different ranges – therefore difficult to compare!

In general, a detailed explanation of the optical/mechanical set-up is lacking, e.g. laser type/wavelength?

Table 1 should be on the same page. I suggest making a plot of the frequency difference error with standard deviation as function of frequency (box plot) for better visualization of results.
Page 7, line 179. Why is e.g. 1 kHz and 1.5 kHz not tested? (I understand that 2 kHz is chosen as threshold frequency, however, the performance at these “lower” frequencies is not documented.)

Table 2: Title: Selected amplitude measurement results
The structure of the table is unclear and not well described. I need more explanation of all rows and columns and in figure 7 indicate the two sigmas, 1.63 and 1.31 with a circle or similar in order to connect table and figure. And in the updated figure 7 write that the two circles indicate the two sigmas from table 2….

Page 209 line 208. The fitted equation y=26.29sin()…..: This can be presented more elegant and scientific correct and what is x – symbol not described. Would it not be possible to use e.g. Hilbert transform for envelope detection?
Figure 8. Figure text: write what the amplitude depth is in the experiment.

Conclusion
It is from the conclusion not clear which improvement that has been specifically obtained in this work compared with previous/others work.
What is the signal to noise S/N ratio of the tested system?
Page 9 line 232: What filter performance are discussed?

General comments:

I do miss a simulation of the system based on figure 3, with known modulation frequency, noise amplitudes and noise frequency spectrum and of course known signal amplitude and frequency. In this simulation the S/N could also be tested (and compared with experiment). Why is 10 kHz chosen for modulation frequency?
Why is the reference modulation carried out in a water tank and not internally in the interferometer using an optical phase modulator?
What’s the similarities between your interferometer approach and laser Doppler interferometry where a high frequency reference modulated beam is also used?

Author Response

(The authors gave the same response as above.)

Round 2

Reviewer 2 Report

Thanks for the revised manus, which has improved significantly.

Regarging “respons 5”, i suggest that the deviations given in the letter could be placed in an appendix in order to clarify that issue.

Author Response

Dear reviewer:

Thank you very much for your attention and consideration.

We have already added an appendix in the paper.

Best wishes!
